# Pediatric COVID-TB: A Clinical Perspective Based on the Analysis of Three Cases

**DOI:** 10.3390/children10050863

**Published:** 2023-05-12

**Authors:** Fabrizio Leone, Martina Di Giuseppe, Maia De Luca, Laura Cursi, Francesca Ippolita Calo Carducci, Andrzej Krzysztofiak, Sara Chiurchiù, Lorenza Romani, Cristina Russo, Laura Lancella, Stefania Bernardi

**Affiliations:** 1Maternal, Infantile and Urological Sciences Department, Sapienza University of Rome, 00161 Rome, Italy; 2Bambino Gesù Children’s Hospital, IRCCS, Immunology an Infectious Diseases University, 00165 Rome, Italy; 3Unit of Microbiology and Diagnostic Immunology, Bambino Gesù Children’s Hospital, IRCCS, 00165 Rome, Italy

**Keywords:** SARS-CoV-2, COVID-19, COVID-TB, children, tuberculosis, TB, coinfection, pediatric

## Abstract

Coronavirus disease 2019 (COVID-19) and tuberculosis (TB) are currently two major causes of death among infectious diseases. Active tuberculosis and a history of tuberculosis appear to be associated with an increased risk of COVID-19. This coinfection, named COVID-TB, was never described in previously healthy children. We report three cases of pediatric COVID-TB. We describe three girls affected by tuberculosis, who tested positive for SARS-CoV-2. The first patient is a 5-year-old girl who was hospitalized for recurrent TB lymphadenopathy. As she never had any complications related to the concomitant infection with SARS-CoV-2, she received TB treatment. The second case is a 13-year-old patient with a history of pulmonary and splenic tuberculosis. She was admitted to the hospital due to deteriorating respiratory dynamics. She was already undergoing treatment for TB, but in the absence of improvement, she also required treatment for COVID-19. Slowly, the general condition improved until discharge. The last patient, a 10-year-old girl, was hospitalized for supraclavicular swelling. The investigations showed disseminated TB characterized by lung and bone involvement without COVID-19-related complications. She was treated with antitubercular and supportive therapy. Based on the data obtained from the adult population and our small experience, a pediatric patient with COVID-TB infection should be considered potentially at risk of worse clinical outcomes; for this reason, we suggest close observation, careful clinical management, and consideration of targeted anti-SARS-CoV-2 therapies.

## 1. Introduction

The clinical spectrum of Coronavirus disease 2019 (COVID-19) in children is wide, ranging from asymptomatic to life-threatening infections. Most children seem to develop asymptomatic or non-severe infections [1,2].

Compared to adults, pediatric patients also show a more rapid recovery and a better prognosis [3]. However, pediatric subjects with underlying diseases may be more susceptible to developing severe symptoms [2]. Conversely, cases of young and previously healthy patients with severe COVID-19 have been reported, suggesting a potentially specific susceptibility to SARS-CoV-2 in these individuals. Generally, fever and cough are the most frequent symptoms in children with SARS-CoV-2 infection. Myalgia, nasal congestion, coryza, odynophagia, headache, diarrhea, and abdominal pain are also common. Gastrointestinal (GI) symptoms are also described; a systematic review showed rates of vomiting of 4–9%, diarrhea of 6–13%, and abdominal pain in approximately 4% of infected patients [4]. After the COVID-19 pandemic, we have witnessed the emergence of multisystem inflammatory syndrome in children (MIS-C), a rare post-infectious hyperinflammatory disorder associated with SARS-CoV-2 [5]. This syndrome indicates that the symptoms of SARS-CoV-2 infection are closely related to the status of the host; indeed, the treatment and prognosis of these inflammatory diseases have not yet been clarified. In addition to the characteristics of the host, after about 3 years from the beginning of the pandemic, it is also evident how viral mutations could modify SARS-CoV-2 infection. SARS-CoV-2 produced several variants, five of which were identified as variants of concern (VOC) by the World Health Organization (WHO) in recent years as follows: Alpha (B.1.1.7), Beta (B.1.351), Gamma (P.1), Delta (B.1.617.2), and Omicron (B.1.1.529) [6].

The initial symptoms of COVID-19 may mimic other respiratory tract infections, such as tuberculosis (TB) or influenza. Coinfections with common pathogens are frequent, and they could interfere with the clinical course of COVID-19. TB and COVID-19 are currently two major causes of death among infectious diseases. Their main route of transmission is through respiratory droplets; in lung tissue, this coinfection (abbreviated COVID-TB) could result in greater damage. A recent review on COVID-TB suggests that in adults, active tuberculosis and a history of tuberculosis appear to be associated with an increased risk of COVID-19 as an aggravation in the prognosis. [7]. During infection, the innate defense is immediately active to restrict the spread of the virus into the lungs. Recognition of viral compounds induces the response of interferon (IFN) type I and the cytotoxic action of natural killer (NK) lymphocytes in the airways [8]. Zhang et al. described that 3% of these subjects, previously healthy, presented with innate errors of IFN type I, Toll-like receptor 3 (TLR-3), and interferon regulatory factor 7 (IRF-7)-dependent pathway. As a result, these patients showed low serum IFN type I [9]. Disseminated forms of TB in children are the result multifactorial mechanism. Host genetic factors, such as single gene inborn errors of immunity affecting the IL-12/IFN-γ (IFN type II) axis, are described. This circuit plays a critical role in the immune response to mycobacterial infections, especially in Pott’s Disease [10], with an important therapeutical implication, as children with such defects can be treated with recombinant IFN-γ. Although SARS-CoV-2 is counteracted by IFN α/β (type I) pathway, it is likely that a deficiency in the activation of the IFN-γ could result in a worse prognosis in patients with COVID-TB. Some studies have shown a delayed production of IFN due to the presence of Coronaviridae. However, in the late phase, IFN reaches a peak compromising its protective role and inducing immune dysregulation. Late stimulation due to the accumulation of activation factors may be the basis of COVID-19-typical dysregulation reactions [4]. In patients with COVID-TB, a deficiency underlying any interferon pathway could play a key role in the evolution of this pathology.

A recent study assessed IFN-I and IFN-II transcription levels in the peripheral blood of children with COVID-19 and in non-infected controls [11]. Higher expression levels of IFN-I and IFN-II inducible genes were found in children with mild or moderate diseases compared to non-infected controls, while their concentrations have declined in children with severe diseases and with MIS-C. The overregulation of IFN-I and IFN-II in children with mild symptoms, their decline in severe cases or with MIS-C, and positive correlations of their transcription in children with SARS-CoV-2 infection suggest that they may play an important role in the course of infection.

However, the investigation of interferon pathways is not a routine test. The demonstration of how interferon alterations could justify the clinical course of SARS-CoV-2 is complicated and even more so in the case of coinfections. The effects of the Corona pandemic on TB disease were very evident in the first twelve months of COVID-19; during that time, the prevalence of newly diagnosed TB, outpatient visits, and newly diagnosed TB infection were reduced [12]. COVID-TB was never described in previously healthy children. In Italy, pediatric admissions with low priority decreased significantly during the pandemic [13]; these data could explain the low incidence of COVID-TB in the pediatric population. We report three cases of pediatric COVID-TB with differing clinical involvement.

## 2. Case Report, Clinical Features of Pediatric COVID-TB Cases

The first patient is a 5-year-old girl with a history of tubercular lymphadenitis (right laterocervical and submandibular lymph nodes), diagnosed in 2019. At the time of the diagnosis of tuberculosis, her chest radiograph showed no alteration. She had correctly performed the anti-tubercular therapy with Ethambutol, Rifampin, Isoniazid, and Pyrazinamide for the first 2 months and then with just Rifampin and Isoniazid for a further 4 months, with evidence of clinical improvement. Approximately 4 months later, the patient underwent surgery to excise the persistent right submandibular granulomatous lesion, confirming pyogenic tubercular granuloma. Due to another recurrence of right cervical swellings, she was hospitalized again; an MRI of the neck documented the presence of multiple lateral cervical lymph nodes bilaterally increased in size. As a result of the positive SARS-CoV-2 infection confirmed via RT-PCR (Real-Time Polymerase Chain Reaction) nasopharyngeal swab (Figure 1), the patient was transferred to our Department of Infectious Diseases for further treatment.

Due to the presumed reappearance of TB lymphadenitis, TB therapy was re-started with Isoniazid, Rifampicin, Ethambutol, and Pyrazinamide. Histological examination revealed chronic necrotizing granulomatous lymphadenitis, consistent with mycobacterium origin. PCR and culture on lymph node biopsy yielded positive results for Mycobacterium Tuberculosis Complex. During her recovery, routine blood chemistry tests were normal. Immunological examinations showed no alterations (Table 1). After 2 weeks, she repeated RT-PCR nasopharyngeal swab and still tested positive.

During hospitalization, the girl showed a good clinical course; apparently, she did not show any symptoms related to SARS-CoV-2 infection. She resulted negative for RT-PCR nasopharyngeal swab after 3 weeks. The patient had SARS-CoV-2 infection in January 2022; VOC was not identified. However, using the Italian report that described the main data on SARS-CoV-2 variants circulating in Italy during the period from 1 January to 14 February 2022 [18], we hypothesized that the viral variant that affected our patient was the Omicron variant. In fact, during that period, the Omicron variant represented 90% of the performed sequences. Of the three identified lineages (BA.1, BA.1.1., and BA.2), BA.1 was the most frequent.

The second patient was an 11-year-old girl who had already been monitored at our department for pulmonary and splenic tuberculosis; she was first admitted for persistent fever and cough. Thoracic CT showed multiple nodal swellings compressing the respiratory tract (Figure 2) and two hypodense areoles at the upper extremities of the spleen. Specific therapy with Isoniazid, Rifampicin, Pyrazinamide, and Ethambutol was started.

Due to worsening cough and dyspnea, she was hospitalized again approximately three months later and tested positive for SARS-CoV-2 (RT-PCR nasopharyngeal swab). The COVID-19 treatment identified at that time (July 2020) was initiated with subcutaneous enoxaparin, azithromycin, and dexamethasone; specific TB treatment was continued. At that time, the only variant of SARS-CoV-2 was the Alpha variant [18]. Other microbiological investigations were carried out with negative results. The presence of tuberculosis on expectoration (PCR-DNA) was re-affirmed. SARS-CoV-2 was also studied in fecal, urine, and ocular samples using a molecular method, with negative results. She performed a new thoracic CT showing radiological worsening compared to the previous check. In particular, the examination showed widespread lymph adenomegaly, especially in the middle mediastinum and right hilum, characterized by a peripheral ring enhancement for central colliquation, severe compression on the right bronchi, and disease in the medium and lower lobes. Various bronchoscopies were required to groom the occlusive tubercles. Intravenous methylprednisolone therapy was also required to improve the inflammation of the lungs. Another thoracic CT showed a complete obstruction of the middle lobe with atelectasia and an adenobronchial fistula. The patient then underwent further bronchoscopy with the aspiration of the caseous material of the main right bronchus. The latest CT showed complete resolution of the medium lobe atelectasis. The immunological examinations carried out during the stay did not detect any alterations. The clinical condition of the patient gradually improved, and respiratory secretions were reduced. After 2 weeks, her RT-PCR naso-pharyngeal swab for SARS-CoV-2 resulted in negative.

The third patient, a 9-year-old Indian girl, was taken to the emergency room for right supraclavicular swelling. The SARS-CoV-2 via RT-PCR performed by screening was found to be positive, although the patient did not present any symptoms of infection. As a Pt1 patient, this child was also infected with SARS-CoV-2 in January 2022; in this case, the most likely variant was the Omicron variant (BA.1) [18].

She presented a high value of cycle threshold (CT) that was compatible with an early phase of the infection course; she tested negative after 2 weeks (Figure 1). Blood chemistry showed neutrophil leukocytosis and increased PCR (Table 1). The chest X-ray showed a right apical thickening and a widening of the distal epiphysis of the right clavicle. An ultrasound was also performed of the lymph nodes in the neck that showed a reactive appearance.

Test IGRA (Interferon Gamma Release Assay) resulted positive; therefore therapy with Isoniazid was established. PCR research of mycobacteria on the first sputum was found positive for Mycobacterium tuberculosis Complex, so Rifampicin, Pyrazinamide, and Ethambutol were added to the treatment. Chest CT showed bilateral pulmonary densities (Figure 3), mass in the upper mediastinum, and erosion of the vertebral bodies with the suspected invasion of the spine canal. Due to the expansion of the disease, methylprednisolone (2 mg/kg/day) was started. Given the suspicion of spine involvement, she was evaluated by orthopedists, and neurosurgeons recommended the execution of an MRI of the spinal-bone-marrow; the exam showed involvement of the cervical and dorsal rachis, particularly the vertebral metameres between C6 and T2. The spine was then stabilized via immobilization with Halo Vest. The general clinical condition of the patient has gradually improved. 

## 3. Discussion

As previously mentioned, adult patients with COVID-TB are more likely to have a severe illness than patients with COVID-19, so they should be considered at risk of poor prognosis. Data about this coinfection in the pediatric setting is very limited. As discussed in our three case reports, the clinical manifestation appears to be very different. Two of our cases showed a more severe clinical course, probably due to pulmonary involvement. In adults, especially those over 65 years and those with chronic conditions, COVID-19 is associated with worse outcomes. Risk factors include severe comorbidities such as medical complexity requiring mechanical ventilation, impaired mucociliary clearance resulting from neurological conditions, obesity (BMI ≥ 35), severe underlying heart or lung condition, or immunocompromised status. However, pediatric data on serious COVID-19 risk factors are more limited and provide less certainty than adult data. Increased risk of severe pediatric COVID-19 was described in children affected by type 1 diabetes, congenital heart and circulatory abnormalities, and obesity [19]. 

Although our patients had no chronic diseases at the time of diagnosis, it is still essential to consider unrecognized host susceptibilities, such as immunodeficiency conditions and the role of the immunopathogenic mechanisms underlying COVID-TB. Studies of cytokine and lymphocyte subpopulation profiles in children indicate how the immune system responds better than adults to SARS-CoV-2 infection; despite this, children have been experiencing a new multisystem inflammatory syndrome associated with SARS-2 since the beginning of the pandemic [20]. As a result, even though children are less susceptible to COVID-19, some have a completely different risk of illness than adults. Data collected from adults suggest that people with latent or active TB are more susceptible to novel coronavirus [7]. Recently, a case of pediatric COVID-TB has been described in a patient with Down syndrome [21]; the severity of the clinical course, in this case, could be suggested by factors related to proinflammatory cytokines rather than expression levels of angiotensin-converting enzyme (ACE) 2 as in adults. Our patients showed no evidence of immune deficiency (Table 1); in particular, they had a normal distribution of lymphocyte subpopulations and a normal CD4/CD8 ratio. However, a more in-depth immunological study, including the study of interferon, along with its therapeutic use, has not been analyzed. Studying this molecule and its sub-forms is rare and costly. It will therefore be difficult to evaluate whether a particular deficiency or mutation may be the common element of susceptibility or immunopathogenesis of COVID-TB. The lack of awareness of certain deficiencies and the lack of specific therapeutic support, therefore, calls for a more in-depth study of patients with these types of coinfections. Despite the current scarcity of COVID-TB in children, we need to understand the clinical differences between the few patients who have been co-infected. Although we have not analyzed the IFN pathways in our three patients, we believe a methodological study involving more COVID-TB patients could shed light on how these two infections work together. Rapid identification of vulnerable people is also important. Considering the SARS-CoV-2 mutations and their impact on the seriousness of COVID-19, it may be important to understand their role in co-infection with COVID-TB. The first variant of SARS-CoV-2 (alpha) was associated with the most adverse outcome of the disease [22]. Pt2 was the only one affected by the alpha variant; she was the only patient with respiratory symptoms and required specific therapy for SARS-CoV-2. Otherwise, Pt.1 and Pt.3, likely affected by the Omicron variant, had no respiratory symptoms and did not require a specific therapy for SARS-CoV-2. None of the patients were vaccinated against SARS-CoV-2, so the differences in clinical presentation were probably due to the different variants of the virus and previous clinical conditions of patients. We, therefore, think that further and larger studies are necessary to clarify pediatric COVID-TB prognosis. Identifying COVID-TB children as patients with a high risk of severe disease would mean detecting patients who might benefit from early and targeted anti-SARS-CoV-2 therapy. However, the recommendations for the therapeutic management of children are derived from safety and efficacy data from adult clinical trials [23]. Data on the use of monoclonal antibodies or antiviral therapies in pediatric patients with SARS-CoV-2 infection are still very limited [24]; much of the use involves off-label administration of available options. 

Among the antiviral therapies used against SARS-CoV-2 infection, only Remdesivir has received Food and Drug Administration (FDA) approval in hospitalized and non-hospitalized children aged ≥ 28 days and weighing ≥ 3 kg [25]. However, in COVID-TB therapy, it is important to consider the possible interaction between Rifampicin and Remdevisir, specifically, that rifampicin is able to decrease serum concentrations of the active metabolite of Remdesivir, although the clinical importance of this exposure remains unclarified [26]. 

Clarifying the pathogenesis behind COVID-TB and the prognostic characteristics in children will be necessary to identify patients who should be considered for further investigation and targeted therapies. We suggest that health professionals need to consider COVID-TB as a clinical condition that requires close observation and careful clinical management.

## Figures and Tables

**Figure 1 children-10-00863-f001:**
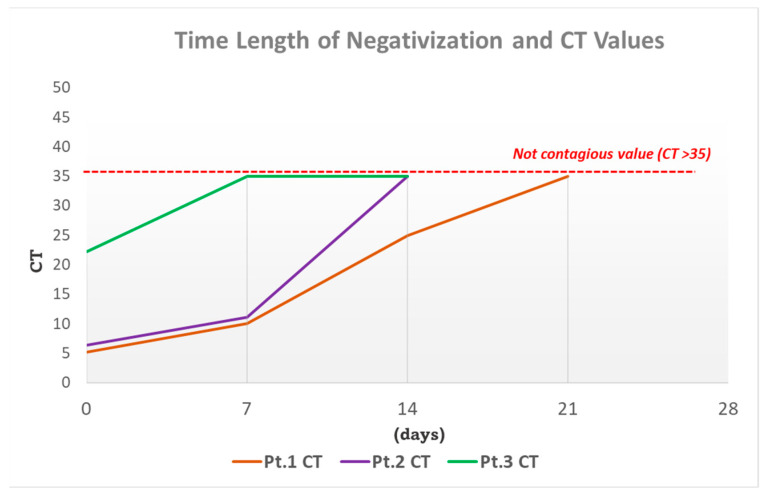
Time length of negativization and cycle threshold (CT) Values. CT values < 20 are indicative of high viral load and CT values ≥ 35 are indicative of low viral load. Positive CT ≥ 35 indicates that SARS-CoV-2 is present in the biological sample in non-infectious form in 95% of cases [14,15]. The patients were positive for E gene, RdRp/s gene, and N gene: the picture reflects the positive trend of CT (N gene) during the days of hospitalization. All three patients had a negative antigen swab in accordance with a CT value ≥ 35.

**Figure 2 children-10-00863-f002:**
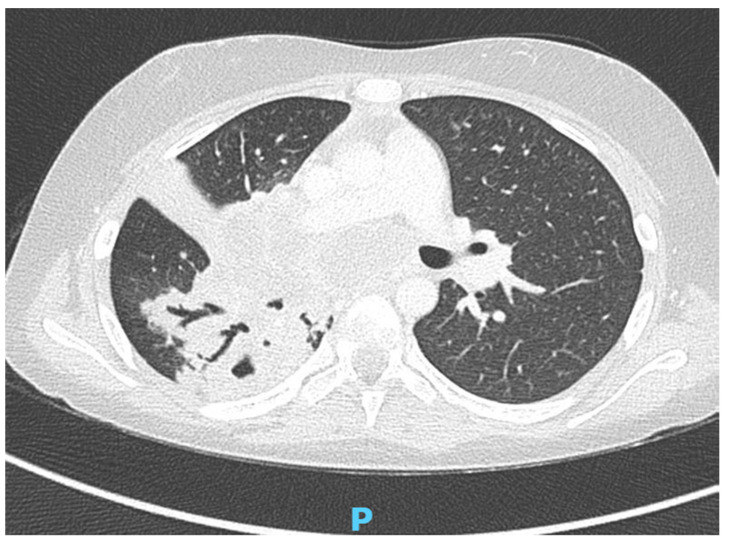
Patient 2’s chest CT showed multiple nodal swellings compressing the airways.

**Figure 3 children-10-00863-f003:**
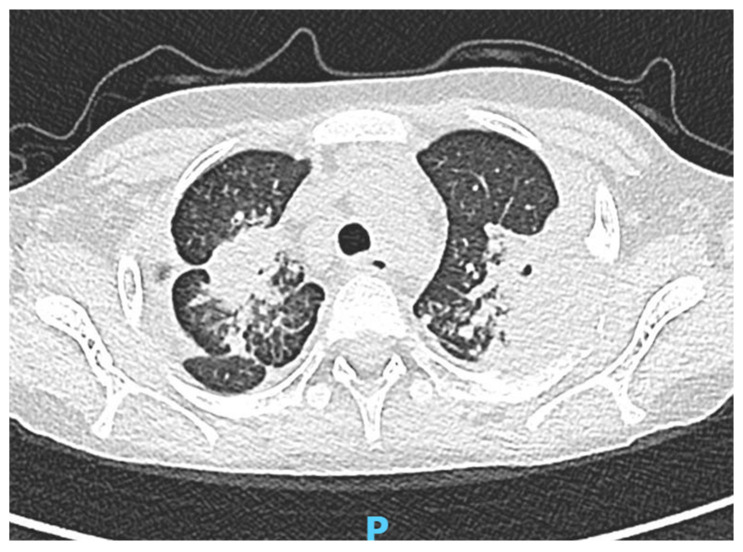
Patient 3’s chest CT showed bilateral pulmonary densities.

**Table 1 children-10-00863-t001:** Clinical and laboratory characteristics of pediatric patients with COVID-TB. * In 2021, there is no specific therapy for SARS-CoV-2 in children; in this case, we used enoxaparin subcutaneously, azithromycin, and Dexamethasone [16,17]. CRP: C-Reactive Protein; VOC: Variant of Concern. ** The variant was identified via the website: https://www.epicentro.iss.it/coronavirus/sars-cov-2-monitoraggio-varianti-rapporti-periodici. Accessed on 2 April 2023.

	Pt.1	Pt. 2	Pt. 3
**Age of SARS-CoV-2 infection**	5 y and 1 m	11 y	9 y
**Age of TB infection**	4 y and 3 m	11 y and 3 m	9 y
**Sex**	F	F	F
**Previous TB**	Yes	Yes	No
**Site**	Lymphadenitis	Lung, mediastinal, and splenic	Lung, Bone (Pott’s Disease)
**Symptoms at admission**	Lymph nodes swelling	Dyspnea	Lymph nodes swelling
**Positive Intradermal Mantoux**	Yes	Yes	Yes
**IGRA**	Yes	Yes	Yes
**Chest CT imaging consistent with TB**	Yes	Yes	No
**Chest CT imaging consistent with SARS-CoV-2 infection**	No	No	No
**TB Therapy**	Isoniazid, Rifampicin, Ethambutol, and Pyrazinamide	Isoniazid, Rifampicin, Ethambutol, and Pyrazinamide	Isoniazid, Rifampicin, Ethambutol, and Pyrazinamide
**Surgery**	Excision of right submandibular granulomatous lesion	None	Orthopedic procedure
**SARS-CoV-2 infection therapy ***	None	Enoxaparin Azithromycin Dexamethasone	None
**O_2_ therapy**	None	None	None
**Day of negativization**	21	14	14
**VOC ** [18]**	Omicron BA.1	Alpha	Omicron BA.1
**Leucocyte count** **(5.5–15 10^3^/uL)**	**5.17**	**4**	**15.71**
**Neutrophil count** **(2–8 10^3^/uL)**	3.54	2.08	12.61
**Lymphocyte count** **(2.2–8.55 10^3^/uL)**	1.14	1.47	2.05
**CRP** **(<0.5 mg/dL)**	0.45	0.38	8
**Immunoglobulins**			
** *IgA* **	Normal	Normal	Normal
** *IgG* **	Normal	Normal	Normal
** *IgM* **	Normal	Normal	Normal
** *Lymphocyte subpopulations* **			
**CD3 Pan T**	62.90%	80.80%	79.00%
**CD4 T Helper**	34.90%	42.80%	39.80%
**CD8 T Suppr/Cytotox**	20.30%	28.30%	30.10%
**CD19 Pan B**	27.70%	10.60%	16.70%
**CD16^+^CD56^+^**	8.40%	7.50%	4.00%
**CD4+/CD8+**	1.7	1.5	1.3

## Data Availability

Not applicable.

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
