# Peer review of "Pediatric COVID-TB: A Clinical Perspective Based on the Analysis of Three Cases"

_children, 2023, doi:10.3390/children10050863_

Round 1

Reviewer 1 Report

Interesting presentation of these cases.

Yet without controls, it is impossible to state that these pediatric TB-Covid coinfections had a significantly different clinical course. 

In no case report the ct-value of the PCR was given. Were these suggestive of an active/recent infection or of an infection that was at the end of its course? 

The table that is presented here with medians and ranges is useless, it has to be replaced with a table that includes the individual values for each case (including all relevant clinical and other values; e.g. year of presentation). 

The introduction does give an interesting overview of some of the relevant literature, but the layout is somewhat confusing. In paragraph 2 the clinical symptoms are discussed, then the pathophysiology in two paragraphs and in the last one a symptom (fever) is discussed. There could be a better build-up of this intro.

Author Response

We thank Reviewer 1 for the valuable corrections.

Certainly, in the absence of healthy controls it is impossible to define a difference in the clinical course: however, currently no pediatric cohorts of COVID-TB are described.

We have modified and improved our table with the required data.

The introduction has been revised accordingly.

Thanks again.

Reviewer 2 Report

Thank you for the opportunity to review the submitted manuscript regarding the coincident occurrence of tuberculosis and COVID-19.

Critical language revision is needed, individual sentences are alienated from their intended meaning.

Please find below my additional comments:

The review contains a description of three cases that apparently occurred at the height of the COVID-19 pandemic, well before the widespread availability of vaccines or the now significantly lower pathogenicity of the prevailing viral variants. Unfortunately, a more detailed specification of which virus variant was determined or suspected does not take place. Regrettably, the data only suggest a secondary diagnosis of COVD-19 and not a relevantly interacting infectious event.

Also the presented CT-images seem to be rather representative regarding the existing TB but not typical for COVID-19, which is certainly in line with the majority of asymptomatic course.

The submitted manuscript is not adherent to the ICMJE recommendations for authorship: >>Substantial contributions to the conception or design of the work; or the acquisition, analysis, or interpretation of data for the work; the work or revising it critically for important intellectual content; Final approval of the version to be published.<<

The statement: "cooperated with the patient's clinical management." does not even remotely correspond to this requirement as fulfillment of the criteria of co-authorship!

It is unclear to the reviewer why the authors resort to a detailed discussion of possible biochemical pathophysiological mechanisms in the introduction of the case presentation. In particular, the long discourse on IFN-α/β/γ pathomechanisms would require appropriate referencing with regard to evidence for the statements on COVID-19 and MIS-C. Besides certainly positioning itself more aptly in the discussion, the present study does not provide any data or new evidence on this issue. How is this supposed to provide relevant information to the reader of this case report?

Table 1 did not include any relevant information that was not already apparent from the case descriptions.

Author Response

We thank Reviewer 2 for the valuable corrections.

We have completely reviewed the quality of English.

We now changed the authorship statement according to the ICMJE recommendations for authorship.

The viral variants are now added into Table 1. Of course the CT images showed just TB infection: in children is very rare to assist to a real COVID-19. In fact the description of IFN mechanism wanted to lead the reader to analyze other effects of SARS-CoV-2: MIS-C is right now the only name we have to call the hyperinflammation after a SARS-CoV-2 in children, but we believe the interactions between multiple agents could be to different clinical course. However we don’t study IFN assays, we want just to stressed this topic for next studies in the future. It’s true that the data only suggest just a secondary diagnosis of SARS-CoV-2 infection, but like we wrote in the manuscript a recent review on COVID-TB suggests that in adult both active TB and a previous history of TB seem to be related to an increased risk for the development of COVID-19, as a worsening of prognosis. So we believe it is important to describe these casae as co-infections: Pt2 was diagnosed for TB just 3 months before SARS-CoV-2 infection.

We modify and improve our table.

Thank you again.

Reviewer 3 Report

Good morning for all authors,

Analyzing with attention and interest the Manuscript (Case report) with ID: children-2303844-peer-review-v2+Fig.1-2 entitled "Pediatric COVID-TB: new co-infections need new strategies" for a possible publication in Journal Children – MDPI (ISSN: 2227–9067; IF=2.835), Section: Pediatric Infectious Diseases,

In conclusion:

I Accept in present form!

Author Response

Thank you so much. We now changed some aspects of our manuscripts. We would be grateful to receive your review also on this new version.

Round 2

Reviewer 1 Report

The authors have adequately addressed my comments and criticism. The manuscript has been substantially improved.

Author Response

Thank you for you answer. We will further check all typos and English Language

Reviewer 2 Report

Unfortunately, the case series is still relatively small, and the authors themselves now disclose that the majority of cases were caused by infection with the much milder, viral variants (Omicron) that are now widespread, and that these were incidental findings in an asymptomatic course. From the reviewer's point of view, this continues to raise considerable doubts:

In order to make statements regarding a relevant co-infection, the sample size should be considerably expanded and, in particular, coninfections with supposedly more aggressive subforms should be investigated (April 2020 - December 2021) in order to be able to make statements about possible renewed aggressive subforms of the SARS-CoV-2 virus.

I explicitly do not share the authors' opinion that such a dedicated publication of patient data should or can take place without an ethics committee approval and, possibly, an ethics waiver. The combination of age, ethnic origin, gender and treating hospital allows direct identification for persons from the patient's personal environment. I therefore urge the authors to provide a consent form from the patient's representative and/or an ethics approval.

Table 1: Still provides redundant information. In addition, to my knowledge, Enoxaparin has never been used as a SARS-CoV-2 specific therapy and there is a spelling error in Dexamethasone.

Author Response

Thank you for your answer and your comments. During the COVID-19 pandemia newly diagnosed TB disease were reduced (10.1183/13993003.01786-2021). This obviously results from the reduced access to hospital by patients which has led to an apparent decrease in  TB diagnosis and controls in patients already diagnosed with TB. In particular, diagnosing COVID-TB was particularly difficult during the Pandemic period when Alpha Variant was the main variant. Nowdays we found ( and we have already mentioned in the manuscript) just a single pediatric case report of COVID-TB (http://www.scielo.br/scielo.php?script=sci_arttext&pid=S1519-38292021000300553&tlng=en) in 2020.  We know that our cohort is really small, however the strong difference between the case infected by the alpha variant leads us to share the idea that in front of new aggressive forms from SARS-CoV-2, a specific therapy for the virus must be undertaken in patients with TB. 

We gave the informed consent signed by the caregivers of the 3 patients during the submission of the paper: in any case we felt it was right to eliminate the ethnic data. The rest of the table contains data that can simplify the view of the case reports.  

Enoxaparin was indicated in COVID-19 in 2020 ( Pediatric PMC9906345, multionational network  10.1136/bmj.n1038). We corrected "Desamethasone" with Dexamethasone

We have inserted in the manuscript the references cited in this answer, modifying small parts of the text.
We have minimally reshaped the table that although redundant allows a quick reading of the cases.
We agree with the auditor on the small number of cases, We have inserted in the manuscript the references cited in this answer, modifying some small parts of the text.
We have minimally reshaped the table that although redundant allows a quick reading of the cases.
We agree with the auditor on the small number of cases, but as explained above the particular situation due to the pandemic can explain this fact.